# Research on the Spatio-Temporal Impacts of Environmental Factors on the Fresh Agricultural Product Supply Chain and the Spatial Differentiation Issue—An Empirical Research on 31 Chinese Provinces

**DOI:** 10.3390/ijerph182212141

**Published:** 2021-11-19

**Authors:** Xuemei Fan, Ziyue Nan, Yuanhang Ma, Yingdan Zhang, Fei Han

**Affiliations:** School of Management, Jilin University, 5988 Renmin Street, Changchun 130022, China; nanzy0918@mails.jlu.edu.cn (Z.N.); mayh21@mails.jlu.edu.cn (Y.M.); zyd20@mails.jlu.edu.cn (Y.Z.); han_fei@jlu.edu.cn (F.H.)

**Keywords:** environmental factors, sustainable development of the fresh agricultural product supply chain, endogenous problems, multi-scale geographic weighted regression (MGWR), China

## Abstract

Environmental factors in time and space play a critical role in advancing the sustainable development of the fresh agricultural product supply chain. This paper, availing the panel data of 31 Chinese provinces from 2008 to 2019, constructs a system of indicators assessing the development of the fresh agricultural product supply chain, and obtains the comprehensive development level in the Entropy Weight Method (EWM). Furthermore, it establishes a comparison between optimal solutions generated by the Instrumental Variables Method (IVM) and the Generalized Method of Moments (GMM) over the endogeneity issue of variables, creates the comparison between the weighted regression methods of Geographically Weighted Regression (GWR) and Multi-scale Geographic Weighted Regression (MGWR), and obtains the relationship among the 14 environmental factors in their spatio-temporal impacts on the development of the fresh agricultural product supply chain. The results indicate that: (1) the environmental influencing factors in this paper have significant endogenous problems and various environmental factors impact on the fresh agricultural product supply chain in different trends and to different degrees. (2) With different bandwidths, the environmental factors could impact the fresh agricultural product supply chain to greatly varied degrees, demonstrating a strong attribute of regional correlation.

## 1. Introduction

In recent years, fresh agricultural products have been taking an increasingly larger proportion of people’s daily consumption. The distribution of these products not only concerns the subsistence of hundreds of millions of farmers, but also bears on the rights and interests of consumers, making it a highly topical socialization issue [1]. The performance of the food supply chain, including food security and nutrition, is subject to the influence of various factors, such as the climate, economy and human activities [2]. On this basis, the fresh agricultural product supply chain (FAPSC) faces greater uncertainty in value and quality [3]. In their transportation, some inevitable factors can result in product damage and quality loss, bringing forth necessary resource consumption and thus higher costs [4]. In 2019, the total output of agricultural products in China amounted to 1.98 billion tons, among which over 1.1 billion tons were fresh agricultural products, which has brought grave pressure on the current supply chain, and constituted one of the important reasons for the high damage and decay ratio of fresh agricultural products [5]. In addition, regions develop the industry of fruit and vegetable planting, poultry breeding, cattle and sheep feeding or other industries in the light of local conditions, which results in certain differences in the production distribution of fresh agricultural products [6]. In this sense, the characteristics of fresh agricultural products determine the uncertainty and vulnerability of its supply chain. Building a sustainable supply model and relieving and reducing its negative influence is a necessary way to form a green and sustainable development model. However, in the prolonged development process, the supply chain of fresh agricultural products is affected by various factors that hinder development and cause serious losses. In order to achieve the goal of sustainable development, it is very important to effectively control these influencing factors.

The reduction in the risks of the whole FAPSC is of high importance. There has been lots of research on the risks of the FAPSC. By constructing a Bayesian network model, two scholars, Juan Yang and Haorui Liu, analyzed the vulnerability of each link of the supply chain and calculate the probability of risk occurrence [1]. Xiaoyu Bian researches and identifies social and natural risk factors that may produce adverse impacts on the upstream of the FAPSC and holds that for further development of the supply chain, an effective risk monitoring mechanism of fresh agricultural product supply should be established to mitigate the adverse impacts produced as a result of the hierarchically progressed positive correlation [7]. Yangang Feng, together with other scholars, probe into the influence relationship among product freshness, prices at the sales end, and the upstream and downstream of the FAPSC, on which basis they constructed a decision-making model for the FAPSC composed of suppliers and retailers, proving that low freshness and high prices of fresh agricultural products can not only impact on demand, but also induce supply risks for suppliers [6]. Jinghan Zhang et al., starting from the transportation of fresh agricultural products and focusing on the problem of “supply chain disruption” in the distribution of fresh agricultural products, evaluate the uncertainties and reliability that may affect cold chain distribution systems with the Fault Tree Analysis and Bayesian network model [8].

Sustainable development has always been the pursuit of the development of FAPSC. Baofeng and another two scholars conclude that, in the mode of reciprocal co-existence, the relation between upstream and downstream of the FAPSC is stable and sustainable [3]. Cristiana designs a type of qualitative indicator monitoring tool for the FAPSC and verifies the feasibility of sustainability assessment through a long-standing method that is user-friendly, highly interactive, and has a high degree of engagement [9]. Sachin et al. use the integrated framework of Interpretative Structural Modelling (ISM) and the fuzzy Decision-Making Trial and Evaluation Laboratory (DMTAEL) and select and analyze 10 potential drivers that may affect the sustainability of the agricultural product supply chain [10]. Jingjie Wang, proceeding from packaging along the FAPSC, proposes that packaging materials with stronger sustainability characteristics can enhance food security and packaging sustainability of fresh agricultural products, and help reduce the overall costs of the fresh agricultural product chain [11]. Dan Han et al. carried out research on the perception of downstream consumers about supply chain risks, finding that the manifestation of the consumers’ purchasing intention can reflect the stability and efficiency of the supply chain on the whole [12].

Finding out the factors affecting the development of FAPSC helps with adopting appropriate measures as the situation requires. Aramyan et al. hold that environmental changes, such as temperature and humidity, can be reflected in the quantity and quality of fresh agricultural products, and thus stress supply chain logistics and quality management [13]. Adi Djoko Guritno, by applying the Analytical Hierarchy Process (AHP), proves that the factors of cost and capital have a higher level of importance than others in compliance with the factors determining the performance of the supply chain [14]. Ikhsan Bani Bukhori et al. replace the less representative indicators of the original assessment system with three new indicators, namely supply order fulfillment, product cycle time and supplier lead time [15]. Ranjan Parajuli et al. conclude that the impacts of the same agricultural climate are not uniform for the overall FAPSC; rather, they are largely varied according to plating spaces and methods [16]. Angelina et al. believe that a healthy diet depends on the daily intake of vegetables, which further exerts cyclical impact on the overall supply chain [17].

With a summary of the outcomes of previous research, it can be inferred that scholars have carried out rather abundant research on the FAPSC from various perspectives. Among these studies, a lot are conducted on the factors that can influence the development of the FAPSC. Scholars use some quantitative analysis methods to establish supply chain development assessment systems from multiple dimensions and analyze the influence of natural, economic and social environmental factors on them. However, as the provinces differ in environmental factors, adding that the factors change with time, the FAPSC is subject to varying degrees of influence. On the whole, available research on the impact of environmental factors on the development of the FAPSC across space and time remains insufficient, especially large-scale regional research.

Accordingly, this paper takes 31 provincial administrative regions in China (including provinces, municipalities and autonomous regions, hereinafter referred to as the provinces) as the research objects to probe into the spatio-temporal impacts of environmental factors on the development of the FAPSC. This paper, availing the panel data of 31 Chinese provinces from 2008 to 2019, constructs a system of indicators assessing the development of the FAPSC and obtains the comprehensive development level in the EWM. Furthermore, it avails four model methods, namely the IVM, the GMM, the GWR and the MGWR, to analyze the relationship among the impacts of the natural, economic, and social environment on various indicators of the upstream, mid-stream and downstream of the FAPSC. The research is conducive for the 31 provinces to enhancing the development of their FAPSC according to local conditions. This paper is of significance mainly in that it provides a theoretical basis for the coordinated development between environment and the FAPSC and for the differentiated, sustainable development among regions.

## 2. Model Methods

In the research on influencing factors of the development of the FAPSC, this paper uses the four model methods of IVM, GMM, GWR and MGWR. Each method has its own strengths and weaknesses, and a combined use of them can generate more objective, reliable results. The IVM can effectively solve the problem of endogeneity. Using instrumental variables, the researchers divide endogenous variables into two parts and arrive at consistent estimators with the part that has no correlation with disturbance terms. This method is the most efficient under the assumption of the existence of spherical disturbance [18]. Nevertheless, provided that the disturbance terms demonstrate heteroscedasticity or self-correlation, it is important that the GMM be applied. The GMM allows for the existence of heteroscedasticity or self-correlation in stochastic disturbance terms, and helps the researchers obtain parameter estimators that are more effective than those obtained with other parameter estimation methods [19]. By constructing regional regression equations at each point within the scope of space, the GWR explores the spatial variations of the research subject under a certain bandwidth. Since it takes the regional influence of spatial objects into consideration, the GWR distinguishes itself with a higher level of accuracy. Furthermore, the MGWR, representing an approach improved on the basis of the GWR, allows each variable to have their own level of spatial smoothness and thus addresses the deficiency of a classical GWR model. Additionally, the specific bandwidth of each variable can be sued as the indicator of the spatial scale in the process. Still, the approach featuring multiple bandwidths helps produce the spatial process model that is more real and practical [20].

### 2.1. Instrumental Variables Method (IVM)

IVM is commonly used in solving the endogenous problems. In the research of this paper on influencing factors of the development of the FAPSC, some explanatory variables might have a correlation with disturbance terms. In this case, the IVM works through two-stage least squares. In the first stage, explanatory variables with disturbance terms are divided into two parts—the fitted values of explanatory variables and those related to disturbance terms. In the second stage, the fitted values of explanatory variables are regressed on the explained variables, so as to obtain the regression line of explanatory variables without endogeneity, which is of great importance to empirical research. The model formula is as follows:(1)$$Z_{i}=\beta_{0}+\beta_{1}X_{i}+\varepsilon_{i}$$
(2)$$Y_{i}=\beta_{0}+\beta_{1}X_{i}+\beta_{2}X_{i}+\varepsilon_{i}$$
(3)$$Z_{i}={\hat{\beta }}_{0}+{\hat{\beta }}_{1}X_{i}$$

The final equation is: (4)$$Y_{i}=\beta_{0}+\beta_{1}X_{i}+\beta_{2}\left({\hat{\beta }}_{0}+{\hat{\beta }}_{1}X_{i}\right)+\varepsilon_{i}$$
where $Z_{i}$ and $X_{i}$ are two types of variables, with $X_{i}$ representing mutually independent variables, $Z_{i}$ representing independent variables with a correlation with $X_{i}$, and $\varepsilon_{i}$ as a disturbance term.

In the above-mentioned formula, the estimated values within parentheses compose a preliminary equation, which in essence, are results estimated with an independent uncorrelated variable when some independent variables are affected by other variables. Those outside the parentheses are structural equations.

### 2.2. Generalized Method of Moments (GMM)

GMM is a general method of moment estimation that features advantages that are not available to the IVM. For instance, being capable of handling both linear and non-linear data, the GMM frees researchers from any concerns about certain assumptions of stochastic error terms and allows heteroscedasticity and serial correlation for such terms. The basic idea of GMM is to derive the population moment using the system of equations constituted of sample moment information, and thus the minimum distance estimators under incremental conditions. The formula is as follows:(5)$$p={\bar{m}}^{\prime }q\bar{m}$$
where $\bar{m}$ is the *n*-dimensional matrix vector; *q* represents weighting matrix; and the circumstance where the inverse matrix of the covariance matrix of $\bar{m}$ sample moments equals *q* is the sufficient condition for an effective estimation of *θ*.

Suppose the regression equation as
(6)$$y_{t}={x_{t}}^{\prime }\beta +u_{t}t=1,2,3,\cdots ,n\;;$$
where $y_{t}$ represents the explained vector; ${x_{t}}^{\prime }$ is the exploratory vector; β is the parameter vector; and n is the number of samples.

As the exploratory vector $x_{t}$ might have a correlation with the stochastic error term $u_{t}$, it is necessary to assume that there are vectors containing *n* (*n* ≥ *k*) components, where *k* represents the number of dimensions of the single-equation parameter vector β, and that $Z_{t}$ has no correlation with the stochastic error term. Hence, at time *t*, the vector $Z_{t}$
with *n* variables and $u_{t}$ met the matrix condition of *n*: *E($z_{t}u_{t}$)* = 0.

Based on the above analysis, the formula of sample moments could be expressed as
(7)$$\bar{m}=\frac{1}{n}Z^{\prime }\hat{u}\left(b\right)$$
where $Z^{\prime }$ represents the data matrix of instrument variables and $\hat{u}\left(b\right)$ the residual sequence.

Choosing b as the parameter estimator could minimize the weighted distance of $p=\frac{1}{n^{2}}\left[{\hat{u}}^{\prime }\left(b\right)Z\right]q\left[Z^{\prime }\hat{u}\left(b\right)\right]$. The covariance matrix of the sample moment $\bar{m}$ is expressed as $\Omega =\frac{1}{n^{2}}Z^{\prime }\mathrm{cov}\left(\hat{u},{\hat{u}}^{\prime }\right)Z,$ where Ω, the covariance matrix of the sample moment $\bar{m}$, could be obtained with the White Test of heteroscedasticity, in which sense $q=\Omega^{-1}$.

### 2.3. Multi-Scale Geographic Weighted Regression (MGWR)

The problem of spatial heterogeneity is easily neglected in empirical research on the multi-factor influence mechanism. Though the traditional GWR model could solve the problem to some extent, it is somewhat limited in the processing of the scale of heterogeneity. Compared with the traditional GWR model, the MGWR model takes differentiated heterogeneity scales of coefficients into consideration and attaches different bandwidth to different variations, thus effectively capturing the influence sphere in space, and adapting to the spatial analysis under different positional relationships. Additionally, the model boasts a better performance in data fitting and explanation [21]. The model formula is as follows:(8)$$c_{i}=\sum_{j=1}^{m}\beta_{bwj}\left(u_{i},v_{i}\right)a_{ij}+\varepsilon_{i}\;$$
where c is the explained variable; a is the explanatory variable; i represents the sample area; u and v represent the coordinates of sample areas; $\beta_{\mathrm{bwj}}$ indicates the bandwidth used for the regression coefficient of the jth variable; and $\varepsilon_{i}$ symbolized stochastic disturbance terms.

In the traditional GWR model, all variables share the same bandwidth, yet its several typical kernel functions and bandwidth rules could still be found in the MGWR model. This paper chooses to use the quadratic kernel and the corrected Akaike information criterion (AICc). While traditional GWR models use the weighted least square method in estimation, the estimation method used by the MGWR model is somewhat different, which could be viewed as a Generalized Additive Model (GAM) with the following formula:(9)$$c=\sum_{j=1}^{m}f_{j}+\varepsilon \;$$
where $f_{j}=\beta_{bwj}a_{j}$

This paper applies traditional GWR estimation in the initial estimation of various coefficients used in the MGWR model, after which the residual could be obtained by subtracting true values and initially estimated predicted values. The formula is as follows:(10)$$\hat{\varepsilon }=c-\sum_{j=1}^{m}{\hat{f}}_{j}$$

A traditional GWR regression is conducted with the independent variable $a_{j}$ and the residue $\hat{\varepsilon }$ added up with each additive term ${\hat{f}}_{j}$, from which the optimal bandwidth $bw_{j}$, a new estimated parameter ${\hat{f}}_{j}$ and a new residual $\hat{\varepsilon }$ are obtained to replace previous estimations. This process stops when the parameter estimation of the number m independent variable $a_{m}$ (m < j) converges. This paper performs the convergence testing in a more relaxed way and adopts the convergence criterion with a classical proportion of changes in the residual sum of squares.
(11)$$SOC_{RSS}=\left|\frac{RSS_{new}-RSS_{old}}{RSS_{new}}\right|$$
where $RSS_{old}$ indicates the residual sum of squares in the former step and $RSS_{new}$ represents the residual sum of squares of the current step.

As there are many influencing factors, the various factors and disturbance terms might have a certain level of correlation, yet this could be effectively addressed by the IVM. The great randomness and uncontrollability of the environment make it very hard to obtain precise information about the distribution of stochastic disturbance terms. Nevertheless, the GMM could arrive at more efficient and robust results without such information [20]. By constructing regional regression equations at each point within the scope of space, the GWR explores the spatial variations and relevant driving factors of the research subject under a certain bandwidth, based on which it effectively solves the problem of spatial heterogeneity, only with certain limitations to the processing of bandwidth. In contrary, the MGWR allows each variable to have their own level of spatial smoothness so as to better capture the influence sphere of the space with a better capacity of data interpretation [22]. As the above-mentioned four methods have complementary advantages, the researchers make a combined use of them in this paper.

## 3. Indicator Selection and Data Interpretation

### 3.1. Indicator Selection

#### 3.1.1. Development Assessment Indicators for the FAPSC

Though research on the FAPSC is abundant both at home and abroad, the system of developing the assessment indicators used for the FAPSC remains inconsistent. For the purpose of conducting both vertical and horizontal comparisons of the development of the FAPSC, this paper selects eight indicators to assess supply chain development, namely the output of fresh agricultural products [23,24,25], the Producer Price Index (PPI) of agricultural products [26,27], the value addition of investment in logistics-related fixed assets [28,29], the transaction volume of fresh agricultural products [26,30], the number of booths in the fresh agricultural products market [25,30], the number of multi-functional markets selling fresh agricultural products [25,30], the average retail price index of fresh agricultural products [24,26], and the average Consumer Price Index (CPI) of fresh agricultural products [31]. As shown in Table 1, the eight indicators are categorized based on the upstream, mid-stream and downstream of the supply chain.

The outputs of fresh agricultural products and the PPI of agricultural products are upstream indicators. The former denotes the aggregate output of all kinds of fresh agricultural products and determines the “input value” of the whole FAPSC, so is hence an indicator of supply capacity. The latter refers to the ratio of the first sales price of the current year to that of the previous year, representing fluctuations in the prices of agricultural products sold by producers and reflecting the costs accumulated at the front end of the supply chain.

Mid-stream indicators include the value addition of investment in logistics-related fixed assets, the transaction volume of fresh agricultural products, the number of booths in the fresh agricultural product market, and the number of multi-functional markets selling fresh agricultural products. Data about the value-added of investment in logistics-related fixed assets could not be obtained directly. Nevertheless, based on relevant research, the value addition of the transportation, storage and postal service industry accounts for over 85% of the aggregate value addition of the logistics sector; this could be taken as the value addition of the investment in logistics-related fixed assets. The other three indicators, namely the transaction volume of fresh agricultural products, the number of booths in the fresh agricultural product market, and the number of multi-functional markets selling fresh agricultural products, are all added-up index values of various fresh agricultural products that could reflect bearing capacity of the supply chain and extensibility at the stage of distribution.

The average retail price index of fresh agricultural products and the average CPI of fresh agricultural products fall into the category of downstream indicators. The former indicates the average value of the ratios of the retail price of various fresh agricultural products in the current year to that of the previous year and represents average changes in the retail prices of various fresh agricultural products, and thus reflects the stability of the supply chain. The latter refers to the average value of the ratios of the consumer price of various fresh agricultural products in the current year to that of the previous year, which could reflect the changes in the power of money in purchasing various fresh agricultural products from the perspective of consumers, and thus reveal the efficiency of the supply chain management.

#### 3.1.2. Indicators of Environmental Factors

The supply chain of fresh agricultural products features a wide coverage and a complex structure with many participants, hence the variety of influencing factors. This paper carries out analysis from the three perspectives of the social environment, economic environment and natural environment and selects 14 indicators to probe into the factors that could influence the development of the FAPSC. As revealed in Table 2, the factors of social environment include the number of Internet users (NIU) [32], the number of employed persons in the primary industry (NEPPI) [33], the total sown areas of farm crops (TSAFC) [34], population (POP) [35,36], and the total number of chain retail enterprises (TNCRE) [36]; factors of economic environment are composed of GDP per capita (GDPPC), the value addition of primary industry (VAPI) [33,36], the investment in fixed assets of agriculture, forestry, animal husbandry and fishery (IAFAHF) [32,33], and the per capita disposable income of urban households (FCDIUH) [37]; and factors of natural environment encompass average temperature (AT) [38,39], average relative humidity (ARH) [39], precipitation (PRE) [39], sunshine hours (SH) [40], and total water resources (TWR) [40,41].

Social environment

The first factor of the social environment is the number of Internet users (NIU). In one aspect, NIU reflects the development of the informatized, automatic and intelligent supply chain, and in the other, it indirectly reveals the extension of the sales channel of fresh agricultural products. The second is the number of employed persons in the primary industry (NEPPI), which has direct influences on the production, delivery, transportation, storage and other links of fresh agricultural products, and then impacts on the development of the overall supply chain. Additionally, the NEPPI mirrors the strength of national and local government support by enacting preferential policies. The third is the total sown areas of farm crops (TSAFC). As fresh agricultural products are produced in agriculture, the TSAFC determines supply capacity to some extent. The fourth is population (POP), as not only production but also consumption of all kinds of resources and products needs to be completed by mankind. The population level in a given area and population changes bear a lot on the supply and demand of fresh agricultural products. In terms of population calculation, this paper uses the permanent resident population. The fifth is the total number of chain retail enterprises (TNCRE), which, to some extent, reflects the demand and degree of preference for certain products.

2.Economic environment

The first factor of economic environment is GDP per capita (GDPPC). This indicator could reveal not only the development level of the local economy, but also the livelihood of the local people, their spending power and so forth, and hence its impact on multiple links of the FAPSC. The second is the value-added of primary industry (VAPI). The production, processing, circulation, storage, consumption and other aspects of fresh agricultural products expand to cover three industries. By VAPI, it denotes the balance of subtracting consumption from production of the primary industry in the previous period and serves as an important indicator of operating results. The third is the investment in fixed assets of agriculture, forestry, animal husbandry and fishery (IAFAHF). The amount of fixed asset investment is a money-based indicator for the activities of constructing and purchasing fixed assets. As the FAPSC involves many areas with various types of fixed asset investment, its development is subject to the impact of IAFAHF. The fourth is the per capita disposable income of urban households (FCDIUH). Per capita disposable income refers to the portion of income that the residents could use at their will. With the improvement of people’s livelihood, they set higher standards for the quality and delivery efficiency of fresh agricultural products, which is mainly shown at the consumer end of the supply chain.

3.Natural environment

The first factor of natural environment is average temperature (AT). The annual average temperature of the central cities of each province selected in this paper represents the annual average temperature of the province, and temperature impacts on each and every link related to fresh agricultural products, including production, transportation and storage. The second is average relative humidity (ARH). The relative humidity has certain influence on soil, crop output, and product freshness in the process of storage. The third is precipitation (PRE), which affects not only crop output but also human activities. The fourth is sunshine hours (SH), the length of which changes the growth, sweetness, texture, output, and other facets of crops, thus exerting impact on the whole supply chain. Last is the indicator of total water resources (TWR). Unevenly distributed in China, water resources could affect the type, output, and quality of crops and thus prices and sales volume.

### 3.2. Data Interpretation

The data of this paper were collected from the China Statistical Yearbook, China Logistics Yearbook, China Rural Statistical Yearbook, and Statistical Yearbook of China Commodity Exchange Market from 2009 to 2020, released by the National Bureau of Statistics of China, as well as statistical yearbooks of the provinces, autonomous regions and municipalities. With the vast territory of China, different regions evidently have different levels of social development and different endowments in agricultural resources, and hence divergent natural, social and economic environments. This paper analyzes the relationship of environmental factors and their spatio-temporal impacts on the development of the FAPSC. The time frame extends from 2008 to 2019, and the research covers 31 Chinese provinces in mainland China.

## 4. Empirical Analysis

### 4.1. Analysis of the Development of China’s FAPSC

#### 4.1.1. Weight Calculation for Indicators of the Development of China’s FAPSC

The supply chain of fresh agricultural products features a complex structure with multiple links. Since different indicators have differently important roles to play in supply chain development, calculating the weighting coefficients of various indicators is prerequisite [42]. This paper combines Stata with the Entropy Weight Method (EWM) to determine specific weighting coefficients, and the smaller the entropy, the greater the weighting coefficient of an indicator [43]. To make the panel data from 2008 to 2019 of 31 provinces more comparable, this paper further introduces the variable of time on the basis of two original variables, namely research objects and assessment indicators.

Construct a matrix of original assessment indicators: given that there are *t* years, *n* provinces, and *m* assessment indicators, the value of *t*, *n*, and *m* is 12, 31 and 8, respectively, in this paper. The numerical matrix is stated as $X={\left\{x_{\theta ij}\right\}}_{t*m*n}$, where $x_{\theta ij}$ represents the *j*th indicator of the *i*th object in year θ.Standardize various indicators of the indicator system, during which process $x_{\max}$ represents the maximum value of the indicator *j* and $x_{\min}$ indicates the minimum value of *j*.

Standardization of positive directional indicators: (12)$$x_{\theta ij}{}^{*}=\frac{x_{\theta ij}-x_{\min}}{x_{\max}-x_{\min}}$$

Standardization of negative directional indicators: (13)$$x_{\theta ij}{}^{*}=\frac{x_{\max}-x_{\theta ij}}{x_{\max}-x_{\min}}$$

3.Calculate the weight of each indicator. The weight of the *i*th object of the *j*th indicator in the indicator is calculated with the following equation:(14)$$Q_{\theta ij}=\frac{x_{\theta ij}{}^{*}}{\sum_{\theta =1}^{t}\sum_{i=1}^{m}x_{\theta ij}{}^{*}}$$4.Calculate the entropy of the *j*th indicator:(15)$$P_{j}=-k\sum_{\theta =1}^{t}\sum_{i=1}^{m}x_{\theta ij}{}^{*}\ln x_{\theta ij}{}^{*}$$
(16)$$k=\frac{1}{\ln(tn)}$$5.Calculate the redundancy rate of each indicator’s entropy. As can be inferred from the above formula, given a specific value of the indicator *j*, the smaller the difference in $x_{\theta ij}{}^{*}$, the larger the value of $P_{j}$, and the greater the difference in $x_{\theta ij}{}^{*}$, the smaller the value of $P_{j}$. On this basis, the redundancy rate of the indicator’s entropy is defined as follows, with a higher redundancy rate indicating greater importance of the indicator in the comprehensive assessment system:(17)$$R_{j}=1-P_{j}$$6.Calculate weighting coefficients:(18)$$V_{j}=\frac{R_{j}}{\Sigma_{j=1}^{m}R_{j}}$$7.Calculate the composite score of the *i*th object:(19)$$S=\sum_{j=1}^{m}W_{j}x_{\theta ij}{}^{*}$$

The weights of indicators calculated with the EWM are as shown in Table 3.

#### 4.1.2. Analysis of the Development of China’s FAPSC

As the socialism with Chinese characteristics has crossed the threshold into a new era, the Chinese population has grown increasingly demanding for food, which makes it more urgent to shatter the traditional operation model of supply chains and proactively promote the integrated innovation of the FAPSC. In recent years, due to the rapid development of the Internet, info-technologies, big data and other new technologies, as well as the continuous innovation in retail models, the FAPSC has grown by leaps and bounds [23].

This paper probes into the development of FAPSCs in 31 Chinese provinces. Being varied in local conditions, different provinces have different types, quality, prices, and time to market of fresh agricultural products, and also uneven situation in terms of infrastructure construction and asset investment. On this account, the development level of the FAPSC varies across provinces. As the data used in this paper span 12 years from 2008 to 2019, the researchers use the average value of each province’s data as the development indicator for further analysis. As shown in Figure 1 and Table 4, the FAPSCs in the 31 provinces are generally well developed.

At the provincial level, Hebei Province, Jiangsu Province, Zhejiang Province, Shandong Province, Guangdong Province, and Henan Province have a higher level of development of the FAPSC, while Hainan Province, Tibet Autonomous Region, and Qinghai Province are less developed in this regard, and the development levels of other provinces are relatively balanced. Jiangsu Province, Zhejiang Province, and Guangdong Province are economically developed and densely populated with superior geographical locations and advanced science and technology, hence their high development level of the FAPSC. Hebei Province, Shandong Province, and Henan Province, though all inland provinces, boast transport networks that radiate in all directions, especially Henan Province and Shandong Province, hence their position as the major distributors and consumers of agricultural products with high efficiency of FAPSC operation. Located in West China, Tibet Autonomous Region and Qinghai Province have less superior geographical locations than the central and costal part of China. Coupled with their less developed economy, complex weather conditions, high altitudes, scarce natural resources and incomplete infrastructure, fresh agricultural products in these provinces are relatively poorly developed. Located in the southernmost part of China, Hainan Province enjoys a tropical monsoon climate, grows abundant tropical plants, and boasts a developed primary industry. However, since Hainan Province started to develop its agricultural product supply chain at a later time, the system is yet to be improved and the logistics operation model remains unitary and conventional. Therefore, the development of its FAPSC lags behind the others.

By geographical area, as shown in Figure 2, the development level of fresh agricultural products in East China (Shanghai Municipality, Jiangsu Province, Zhejiang Province, Anhui Province, Fujian Province, Jiangxi Province, and Shandong Province) is the highest, with the average value of their indicators hitting 0.376536. Coming next are Central China (Henan Province, Hubei Province, and Hunan Province), North China (Beijing Municipality, Tianjin Municipality, Shanxi Province, Hebei Province, Inner Mongolia Autonomous Region) and South China (Guangdong Province, Guangxi Zhuang Autonomous Region, and Hainan Province), with the average values of their indicators reaching 0.275871, 0.242432, and 0.224134, respectively. The last are Northeast China (Heilongjiang Province, Jilin Province, and Liaoning Province), Southwest China (Sichuan Province, Guizhou Province, Yunnan Province, Chongqing Municipality, and Tibet Autonomous Region), and Northwest China (Shaanxi Province, Gansu Province, Qinghai Province, Ningxia Hui Autonomous Region, and Xinjiang Uygur Autonomous Region), with the average values of their indicators registering at 0.168358, 0.149629, and 0.123828, respectively. These results are basically consistent with the actual situation of different regions, including economic development, natural resources and climate, among others.

### 4.2. Instrumental Variables Method (IVM)

Based on reality, it is possible to conclude that some environmental factors have a bidirectional causal relationship with the development of the FAPSC. For instance, the increase in the number of employed persons in the primary industry (NEPPI) can accelerate supply chain development, and supply chain development, in turn, creates more jobs for the primary industry and increases payrolls. The total number of chain retail enterprises (TNCRE) and the value addition of the primary industry (VAPI) also have a similar interrelationship with the development of the FAPSC. To avoid any measurement bias resulting from the endogeneity of explanatory variables in the model, this paper uses Stata in the IVM.

#### 4.2.1. Model Specifications

Considering various factors that can affect the development of the FAPSC, the researchers specify model specifications at the regional level and explanatory variables as follows:(20)$$\begin{array}{l} FAPSC_{\mathrm{it}}=\beta_{0}+\beta_{1}NIU_{it}+\beta_{2}NEPPI_{it}+\beta_{3}TSAFC_{it}+\beta_{4}TNCRE_{it}+\beta_{5}VAPI_{it}\\ +\beta_{6}PCDIUH_{it}+\beta_{7}ARH_{it}+\beta_{8}PRE_{it}+\beta_{9}SH_{it}+\beta_{10}TWR_{it}+\varepsilon_{it} \end{array}$$

Variables in the model can be referred to in Table 2. $\beta_{i}$ represents variable coefficients; the two subscript labels, namely *i* and *t*, stand for “province” and “year”, respectively; $\beta_{0}$ and ε indicate constant terms and disturbance terms, respectively.

#### 4.2.2. Test for Endogeneity and Selection of Instrumental Variables

Whether any endogenous variables exist in the model needs to be determined before using the IVM. This paper conducts a Hausman test to prove the endogeneity of explanatory variables [44]. The null hypothesis of the Hausman test states that, all explanatory variables are exogenous variables. If the null hypothesis is rejected, it should be concluded that endogenous variables exist; otherwise, if the null hypothesis is accepted, the conclusion is that no endogenous variable exists, in which sense the IVM cannot be used. The test results of the Hausman test are as shown in Table 5:

It can be inferred from the statistics of the test results and the *p*-value, at the significance level of 1%, that the model can reject the null hypothesis of the Hausman test, which means that there are endogenous explanatory variables and the IVM is feasible.

This paper takes population (POP), GDP per capita (GDPPC), the investment in fixed assets of agriculture, forestry, animal husbandry and fishery (IAFAHF), and average temperature (AT) as alternative instrumental variables. In order to evidence the availability of alternative variables, the researchers set endogenous explanatory variables as the number of employed persons in the primary industry (NEPPI), the total number of chain retail enterprises (TNCRE) and the value addition of primary industry (VAPI) and conduct multiple linear regression with the above four alternative instrumental variables in separation.

According to the regression results recorded in Table 6, the results of regressing the variable “number of employed persons in the primary industry (NEPPI)” on “average temperature (AT)” are not significant. This is probably because, with the continuous advances in agricultural planting technologies, the output of fresh agricultural products is rarely affected by such environmental factors as temperature. The regression results of the variables “population (POP)”, “GDP per capita (GDPPC)” and “investment in fixed assets of agriculture, forestry, animal husbandry and fishery (IAFAHF)” have all passed the test at a significance level of 1%. As is revealed, the regression coefficient of POP is the largest and a positive one, which may be explained by the fact that POP expansion leads to an increase in the number of workers in all industries, including the primary industry; the regression coefficient of IAFAHF is relatively large as a positive one, which is probably due to the fact that greater infrastructure input helps create jobs, yet the regression coefficient of GDPPC is negative, maybe because that the higher the GDPPC, the more workers are involved in the secondary and tertiary industries; hence, there are fewer workers in the primary industry.

According to the regression results listed in Table 7, the results of regressing the variable “total number of chain retail enterprises (TNCRE)” on “investment in fixed assets of agriculture, forestry, animal husbandry and fishery (IAFAHF)” are not significant. A possible explanation is that, while IAFAHF represents investment in the upstream of the FAPSC, TNCRE is an indicator of the downstream of supply chain; adding that there are many intermediate links, the driving effects of IAFAHF on TNCRE are not obvious. The regression results of the variables “population (POP)”, “GDP per capita (GDPPC)”, and “average temperature (AT)” are all very significant and have passed the test at a significance level of 1%. To be specific, the regression coefficient of AT is a positive value of 289.0761. Therefore, it can be deemed that AT and TNCRE have rather significant positive relations, maybe because the higher the AT, the less easy it is to preserve fresh agricultural products, leading to a larger number of chain retail enterprises owning refrigerators.

According to the regression results noted in Table 8, the *p*-value of the variable “GDP per capita (GDPPC)” is 0.2065, and that of the variable “investment in fixed assets of agriculture, forestry, animal husbandry and fishery (IAFAHF)” is 0.1507. The results of regressing “value-added of primary industry (VAPI)” on these two variables are both not significant. The reason is likely because that both GDPPC and IAFAHF are economic factors that can increase the output of fresh agricultural products along the supply chain, whereas the “total number of chain retail enterprises (TNCRE)” is largely affected by demand, hence rarely affected by the two alternative instrumental variables. The regression coefficient of “population (POP)” stands at 0.4138 and that of “average temperature (AT)” is −18.1926, both of which have passed the test at a significance level of 1%. “Population (POP)” and “value-added of primary industry (VAPI)” share relatively significant positive relations, maybe because the larger the population, the higher demand for fresh agricultural products; hence, greater VAPI as production is pulled by demand.

All in all, each of the four alternative instrumental variables can exert significant positive or negative impacts on one or several endogenous explanatory variable(s). This evidences that it is feasible to replace three endogenous explanatory variables with the four alternative instrumental variables.

#### 4.2.3. Two-Stage Least Squares

This paper uses the two-stage IVM for regression analysis [45], and the regression results are shown in Table 9.

As the results show, R^2^ produced by the goodness-of-fit test is 0.6629. In this sense, the fitting results are satisfactory, and the explanatory variables can well explain the predicted variables. The result of the significance test of the formula (F-test) registers 727.83, representing a sound significance level [46]. The regression coefficients of the explanatory variables of the “number of Internet users (NIU)”, “number of employed persons in the primary industry (NEPPI)”, “total number of chain retail enterprises (TNCRE)”, “value-added of primary industry (VAPI)”, “precipitation (PRE)”, and “sunshine hours (SH)” are positive, whereas those of other variables are negative. Furthermore, the regression results of NEPPI, TNCRE, “average relative humidity (ARH)”, “precipitation (PRE)”, and “total water resources (TWR)” are all very significant, passing the test at a significance level of 1%. It can thus be inferred that the NEPPI, TNCRE, and PRE share a highly positive correlation with the “development of the FAPSC”; that is, the larger the NEPPI, the larger the TNCRE, and the greater the PRE, the higher the development level of the FAPSC in a given province. In contrast, the ARH and TWR have a highly negative correlation with the “development of the FAPSC”, which is probably attributed to the fact that as moisture and humidity have a great impact on soil, too much moisture and humidity may result in an oxygen deficit for the roots of crops, and hence a smaller output of fresh agricultural products.

### 4.3. Generalized Method of Moments (GMM)

Where disturbance terms demonstrate homoscedasticity, the traditional IVM is also applicable to the situation of just identification and over identification. Yet, in general scenarios, the variances of disturbance terms are different, and the predicted values with a small variance may contain more information (for example, the variances of variables are largely different across provinces) [47]. To take into consideration over identification and heteroscedasticity in a more comprehensive manner, this paper uses Stata in the GMM.

#### 4.3.1. Model Design

On the basis of considering the existence of heteroscedasticity, the GMM is set up as follows:(21)$$FAPSC_{\mathrm{it}}=\beta_{0}+x_{it}\beta_{it}+\varepsilon_{it}$$
(22)$$\varepsilon_{it}=\upsilon_{\mathrm{it}}+\nu_{it}$$
where $FAPSC_{\mathrm{it}}$ represents the development level of the FAPSC; $x_{\mathrm{it}}$ is the vector of endogenous and exogenous explanatory variables of environmental factors; $\beta_{\mathrm{it}}$ is the parameter vector of each explanatory variable; $\beta_{0}$ is the constant term; $\varepsilon_{\mathrm{it}}$ represents disturbance terms, classified into the fixed disturbance term $\upsilon_{\mathrm{it}}$ and the variable disturbance term $\nu_{\mathrm{it}}$; and two subscript labels, namely *i* and *t*, stand for “province” and “year”, respectively.

#### 4.3.2. Optimal GMM Estimation

While using the IVM with population (POP), GDP per capita (GDPPC), the investment in fixed assets of agriculture, forestry, animal husbandry and fishery (IAFAHF), and average temperature (AT) as instrumental variables, the researchers take into account the influence of heteroscedasticity and obtain the results of optimal GMM estimation, as shown in Table 10.

According to the results of GMM regression analysis, taking into account the influence of heteroscedasticity, the goodness-of-fit test produces an R^2^ value of 0.6743, better than the fitting effect of models which only consider homoscedasticity. To be specific, the regression coefficients of the “number of employed persons in the primary industry (NEPPI)”, the “total number of chain retail enterprises (TNCRE)”, “average relative humidity (ARH)”, “precipitation (PRE)”, and “total water resources (TWR)” can still pass the tests at a significance level of 1%. In contrast, the significance of three explanatory variables, namely the “total sown areas of farm crops (TSAFC)”, “value-added of primary industry (VAPI)”, and “PRE” has embraced further improvement.

#### 4.3.3. Weak Instrumental Variable Test

To ensure consistency in linear estimators, the instrumental variables should meet two conditions, i.e., having correlation with endogenous variables and having exogeneity in themselves. In cases of a weak correlation, instrumental variables are considered weak. The information provided by weak instrumental variables is very limited, and existing approaches to first-stage regularization can lead to a large bias [48]. On this account, this paper uses the diagnosis statistics of three endogenous variables for the test, specifying that if the result of the F-statistics exceeds the empirical value of 10, the related instrumental variable is not a weak one. The test results are as shown in Table 11:

As revealed by the test results, the F-statistics of all three endogenous variables exceeds 10, evidencing that the four instrumental variables are not weak instrumental variables. In addition, both the values of R^2^ and adjusted R^2^ in the regression are larger than 0.7, which shows that instrumental variables contribute significantly to the fitting tests, with little damage caused to estimation accuracy.

#### 4.3.4. Over Identification Test

Where the number of instrumental variables is larger than that of endogenous variables, the scenario is considered “over identification.” With three endogenous variables and four instrumental variables, the researchers can adopt 2SLS for the regression; nevertheless, it would be hard to test for the exogeneity of instrumental variables. In the Sargan–Hansen test of over identification, the null hypothesis states that all instrumental variables are exogenous [49]. The test results are as shown in Table 12:

As revealed by the test results, the statistics of the Sargan–Hansen test stands at 1.0917 with a *p*-value of 0.2961. As the null hypothesis cannot be rejected at a significance level of 1%, all instruments are exogenous and effective.

### 4.4. Multi-Scale Geographic Weighted Regression (MGWR)

#### 4.4.1. Model Comparison

Goodness of fit refers to the fitting degree of the fitted curve for observed values and is reflected in the coefficient of determination. This paper adopts Stata to carry out GWR and MGWR. According to Table 13, the coefficient of determination R^2^ of MGWR, standing at 0.916, is greater than that of the traditional GWR. This shows that the MGWR model is more capable of data fitting. AICc, as another indicator measuring model performance, denotes greater fitting capacity at a lower value. The fact that the value of AICc of MGWR is much lower than that of the traditional GWR testifies that the MGWR model has not only a more complex structure, but also higher goodness of fit. The effective number of parameters of the MGWR model is 70.378, demonstrating that with only little data, the MGWR model can produce more authentic, reliable results.

#### 4.4.2. Scale Analysis

Traditional GWR can only reflect the average value of the influencing sphere of various indicators with a consistent bandwidth of 160, whereas the MGWR can give different bandwidths to different indicators and capture the spatial sphere of influence more effectively. As can be inferred from Table 14, the influencing spheres of different indicators are largely varied. To be specific, the number of employed persons in the primary industry (NEPPI), the total sown areas of farm crops (TSAFC), population (POP), the total number of chain retail enterprises (TNCRE), average temperature (AT), and the average relative humidity (ARH) have the same bandwidth of 53, accounting for 14.21% in the total number of samples. They have a limited sphere of influence and relatively greater spatial heterogeneity. The indicators of value addition of primary industry (VAPI), AT, and total water resources (TWR) also have a small bandwidth and greater spatial heterogeneity. The influencing scale of sunshine hours (SH) stands at 161, approximately the same with that of 15 provinces, mainly as a result of latitude differences between North China and South China. The bandwidth measuring the influencing scale of the number of Internet users (NIU) registers 193, approximately the same as that of 15 provinces, accounting for 51.74% of the total number of samples and demonstrating a large influencing sphere and a lower level of spatial heterogeneity. The bandwidth of the influencing scales of the investment in fixed assets of agriculture, forestry, animal husbandry and fishery (IAFAHF) and per capita disposable income of urban households (FCDIUH) reaches 361. This is an overall bandwidth without spatial heterogeneity.

#### 4.4.3. Indicator Analysis

The statistics of the parameter estimates for MGWR are summarized in Table 15. The value of constant terms ranges from −0.278 to 0.067, with the mean value calculated as −0.097 and the standard deviation as 0.097. This shows that, in different regions, different factors can exert different degrees of influence on fresh agricultural products.

In order to observe the distribution of MGWR parameter estimates of each region more directly, this paper uses ArcGIS to draw spatial patterns of the MGWR coefficients of various environmental factors, which are shown in Figure 3.

The number of Internet users (NIU) demonstrates positive influences, which has intimate relations with the increased coverage of the Internet and the increased number of Internet users over the 12 years from 2008 to 2019. Especially, as recent years have witnessed the leapfrog development of such new technologies and models as info-technologies, big data, 5G, online shopping, and community group buying, the NIU shares a positive correlation with the development of the fresh agricultural product chain. As can be seen from Figure 3a, the regions with high coefficient values, ranging from 0.268 to 0.281, are mainly distributed in the southeastern coastal areas and Northeast China, followed by those in the Central China and Southwest China, whereas most northern and northwestern regions have lower values ranging from 0.218 to 0.350.

The number of employed persons in the primary industry (NEPPI), the total sown areas of farm crops (TSAFC), and the value addition of primary industry (VAPI) all exhibit negative influences. Firstly, an excessively great NEPPI will decrease the number of workers employed in the other links, leading to a decline in overall efficiency. Secondly, as the demand for agricultural products is less elastic, where the other factors remain unchanged, an increase in crop yield or harvest can bring down the income of farmers, discourage workers in the primary industry, and thus hinder the development of the FAPSC [50]. According to Figure 3b, the regions with high values of coefficient, ranging from −0.630 to 0.163, are mainly distributed in Northeast China, Central China and South China, followed by those in Southwest China and Northwest China, whereas the Beijing–Tianjin–Hebei Region and the southwestern and northern regions have values lower than −0.971. The spatial differences across regions are significant.

The population has positive influences with relatively large coefficients. The process or consumption of all types of resources and products depends on mankind. This explains the significant influence of population on the FAPSC. As can be inferred from Figure 3c, the regions with high coefficient values, ranging from 1.062 to 5.786, are mainly distributed in Northeast China and eastern coastal areas, which well suits the economic situation of the two regions. To be specific, as Northeast China is confronted with a development lag and serious brain drain, a population increase will greatly affect its FAPSC; in contrast, for the eastern coastal areas with advanced economy and highly dense population, both an increase and decrease in population have great impacts on not only regional economy, but also the development of the FAPSC.

The total number of chain retail enterprises (TNCRE) demonstrates negative influences. According to Figure 3d, the regions with high values of coefficient, ranging from 0.114 to 0.449, are mainly the Beijing–Tianjin–Hebei Region and those located in eastern coastal areas, followed by those in Northwest China and Central China, while the coefficients of Northeast China are at relatively low levels of below −0.811. Chain retail enterprises operate at the consumption end. The market of chain retail enterprises is almost saturated in developed regions, whereas the outlets of such enterprises remain sporadic in less developed regions. The per capita disposable income and purchasing power of consumers are especially important in this regard. As regions with a less developed economy may face deficient consumption, they can encounter such problems as resource waste and inventory backlog [51].

The GDP per capita (GDPPC) signals positive influences. Since different regions have prominent differences in economic development, and the results are characterized by great spatial heterogeneity. Figure 3e shows that the regions with high coefficient values, ranging from 0.273 to 0.287, are mainly distributed in West China and Northeast China, followed by those in Central China, whereas the southeastern coastal areas have low values ranging from 0.269 to 0.270. Provided that the economy of better developed regions posts a growth of 1% and their supply chains are almost fully developed, the supply chains may still embrace slight improvement, whereas if the economy of less developed regions grows by 1%, their supply chains will benefit from relatively great influences in a positive way.

The value addition of primary industry (VAPI) exerts negative influences with great spatial heterogeneity. As is revealed by Figure 3f, the regions with high values of coefficient, namely over −0.498, are mainly distributed in Southwest China, South China and West China, followed by those in East China, Central China and Northeast China, whereas regions of North China have low values of below −0.915. Leaving aside price fluctuations, a greater value of VAPI generally signals an increase in crop yield, in which scenario the synchronous coordination of other links of the supply chain may result in uncoordinated supply chain development. When price fluctuations are taken into consideration, as price hikes in fresh agricultural products may cause the sales volume to decline, the development of the FAPSC may be affected.

The investment in fixed assets of agriculture, forestry, animal husbandry and fishery (IAFAHF) has positive influences with less spatial heterogeneity. According to Figure 3g, the regions with high coefficient values, ranging from 0.0006 to 0.0008, are mainly distributed in Southwest China, followed by those in Northwest China, whereas regions of North China show low values of below 0.0001. The IAFAHF, in one aspect, improves planting or breeding conditions and increases the output of fresh agricultural products, and in the other, helps expand the sales channel, thus removing the “blockage” in supply chains as a result of increased production.

The per capita disposable income of urban households (PCDIUH) produces negative influences with relatively great spatial heterogeneity. Urban residents are the major consumers of fresh agricultural products. As people’s lives get better and better, they now have stricter requirements for supply chains and prefer products of higher quality, better taste and faster delivery. In this sense, their demand for fresh agricultural products will decrease. According to Figure 3h, the regions with high coefficient values, namely above −0.114, are mainly distributed in Northwest China, followed by those in Southwest China and Northeast China, whereas regions in Central China and East China have low values of below −0.118.

The average temperature (AT) and average relative humidity (ARH) present negative influences. This mainly relates to the growing conditions of crops. For example, a too-high temperature and humidity can affect the growth, photosynthesis, respiration and other aspects of crops, and thus crop output and supply chain development as a whole [52]. As can be seen from Figure 3i, Central China and West China are regions with high coefficient values, ranging from 0.025 to 0.217, while North China and East China are those with low coefficient values, which are less than −0.559. In South China where the humidity is already relatively high and the rainfall is abundant, any increase in humidity will negatively affect crop growth; in North China where its dry climate is suitable for crop growth, any increase in humidity will produce generally negative influences on the growth.

The precipitation (PRE) and sunshine hours (SH) both demonstrate positive influences. According to Figure 3j, the regions with high coefficient values, ranging from 0.356 to 1.517, are mainly distributed in Northeast China and Central China, whereas the regions with low values are mainly identified in West China, posting coefficients of less than −0.031. In one aspect, precipitation affects crop growth—greater rainfall in areas lacking in precipitation benefits crop growth, and in the other aspect, precipitation affects people’s production and social activities and thus the development level of supply chain.

The total water resources (TWR) exert positive influences. As is revealed by Figure 3k, the regions with high coefficient values, ranging from 0.669 to 0.921, are mainly distributed in Northeast China and Central China, whereas the regions of West China have low values ranging from 0.056 to 0.106. China’s fresh water resources account for only 6% of the world’s total. Richer water resources are expected to instill vigor into production activities and advance the sound development of the FAPSC [53].

In general, the number of Internet users (NIU), population (POP), total sown areas of farm crops (TSAFC), precipitation (PRE) and total water resources (TWR) exert relatively greater influences on Northeast China; the average relative humidity (ARH), GDP per capita (GDPPC), investment in fixed assets of agriculture, forestry, animal husbandry and fishery (IAFAHF), per capita disposable income of urban households (FCDIUH), and fluctuations in the value-added of primary industry (VAPI) have rather significant impacts on supply chain development in Northwest China; the ARH, IAFAHF, and VAPI have greater impacts on Southwest China; the NIU, POP, total number of chain retail enterprises (TNCRE) and TSAFC produce the greatest impacts on Southeast China; the VAPI, TSAFC and IAFAHF exert rather prominent influences on South China; the POP, PRE, TNCRE and TWR have the greatest impacts on the Beijing–Tianjin–Hebei Region; and the various indicators demonstrate basically balanced influences on Central China.

### 4.5. Model Comparison

Since the various models have their own strengths and weaknesses, this paper combines the software Stata with four model methods to analyze the factors that may affect FAPSCs in 31 Chinese provinces, with more objective and reliable results obtained. Since some explanatory variables and predicted variables have bidirectional causal relationship, the common linear regression method is inappropriate for fitting. Therefore, this paper primarily adopts the IVM to address the endogeneity problem of independent variables and thus effectively reduce deviations. In the case where heteroscedasticity exists, GMM proves more effective as an IVM. This is the second method used in this paper. With GMM, the researchers are also able to test the exogeneity of instrumental variables and further improve the accuracy of model-generated results. Despite the multiple advantages, these two models neglect the issue of spatial heterogeneity. For the purpose of research, this paper selects 31 Chinese provinces, which denotes a large span and the necessity to consider geographical factors. Compared with GWR, MGWR can capture spatial information more effectively by giving different bandwidths to different indicators. For example, the bandwidths of the number of Internet users (NIU), the total number of chain retail enterprises (TNCRE) and the total number of chain retail enterprises (TNCRE) is 193, 53 and 361 respectively, based on which their impacts across regions are analyzed in a targeted manner. Furthermore, the researchers visualize the coefficients of influencing factors with ArcGIS to probe into the degree of influence of different factors in different scales. The results obtained are more robust, reliable, and thus more capable of contributing to the research subject of this paper, namely “research on the spatio-temporal impacts of environmental factors on the supply chain of fresh agricultural products and the spatial differentiation issue.”

## 5. Conclusions and Suggestions

Based on the data from 2008 to 2019 of 31 Chinese provinces, this paper selects 14 indicators from three perspectives of social environment, economic environment and natural environment, and uses the four models of IVM, GMM, GWR and MGWR in combination to carry out empirical research.

Based on the analysis of Section 4, it is concluded that the IVM passes the Hausman test, with the R^2^ produced by the goodness-of-fit test post the regression analysis results standing at 0.6629 and passing the significance test. This represents an effective solution to the issue of the endogeneity of explanatory variables. Taking into account heteroscedasticity, the goodness of fit R^2^ is 0.6743, demonstrating a more satisfactory fitting result. Additionally, it successively passed the weak instrumental variable test and over identification test as an effective solution to the uncertainties of disturbance terms. Based on this, the degrees of influence of environmental factors on the FAPSC of various provinces are obtained. The R^2^ produced by the goodness-of-fit test post the regression using the GWR model registers 0.788 and the AICc stands at 634.428, yet the bandwidths are both 160, indicating an ineffective capture of the influence sphere of space. Then, the MGWR is used as a complement. It turns out the the goodness-of-fit and AICc values are both better than those obtained with the GWR, meaning that the MGWR more effectively captures the influence sphere of effective space in different locations, and that it can more effectively reflect the issue of spatial differentiation. In addition, a diagram of spatial patterns is drawn for the coefficients of MGWR when considering different environmental factors. With their advantages complementing each other, the four models generate results that are more precise, objective and reliable.

In the future, more attention should be attached to the development of fresh agricultural products, and a new round of optimization, integration and innovation is just around the corner, pushing the market to new heights. To contribute to the sustainable development of FAPSCs of different regions in a way that is more coordinated with environmental factors and is well adapted to local conditions, and facilitate the offering of quality services to the public in greater efficiency, the paper hereby proposes the following suggestions:

East China boasts a high development level of FAPSC. To maintain its leading position, the region needs to grasp opportunities to pursue high-quality development and take preemptive measures to efficiently identify and solve conundrums. Northeast China is endowed with abundant agricultural and mineral resources. Under the influences of its geographic location, economic development level, infrastructure, climate and others, the region lags behind others in the development of the FAPSC. Looking into the future, Northeast China needs to appeal to a wider base of talents, intensify infrastructure construction, and increase the Internet availability rate so as to promote the development of the FAPSC. North China, especially the Beijing–Tianjin–Hebei Region, has a large-scale economy and great economic vitality. Nevertheless, with less diversified agricultural products and fewer workers employed in the primary industry, the region needs to pay greater attention to the development of agricultural products and formulate policies to bring more people into the primary industry, expand the total sown areas of farm crops, ramp up investment in fixed assets, and so forth. Central China, enjoying an advantageous geographic location, developed transportation, abundant agricultural products, as well as a prospering agricultural product market, should not only optimize the structure of its supply chain for greater operation efficiency, but also proactively integrate advances in new technologies into the supply chain to achieve informatized, intelligent and sustainable development. With a unique geographic location, developed waterways and diversified agricultural products, South China also has the problem of few workers being employed in the primary industry and limited emphasis placed on the development of agricultural products. In the future, it is essential that South China increases the number of employees in the primary industry, expand the sown area of crops, and strengthen infrastructure construction in sectors of agriculture, forestry, animal husbandry and fishery. Except for a few provinces and cities, most areas of Southwest China are less economically developed. Southwest China also has a disadvantageous geographic location and terrain structure and less favorable climate conditions. To further attract talent and develop special agricultural products in the light of local conditions, the region needs to make full use of the Internet, e-commerce and other factors to tackle problems in the production, processing, distribution, sales and other links of the supply chain. As for Northwest China, where the FAPSC is poorly developed, the level of informatization remains low, the cost of supply chain runs high, and inefficient operation consumes lots of energy; it is suggested that the region should further optimize the model of its supply chain and step up the input of financial, material and human resources, striving to realize the goal of lowering costs and raising efficiency.

## Figures and Tables

**Figure 1 ijerph-18-12141-f001:**
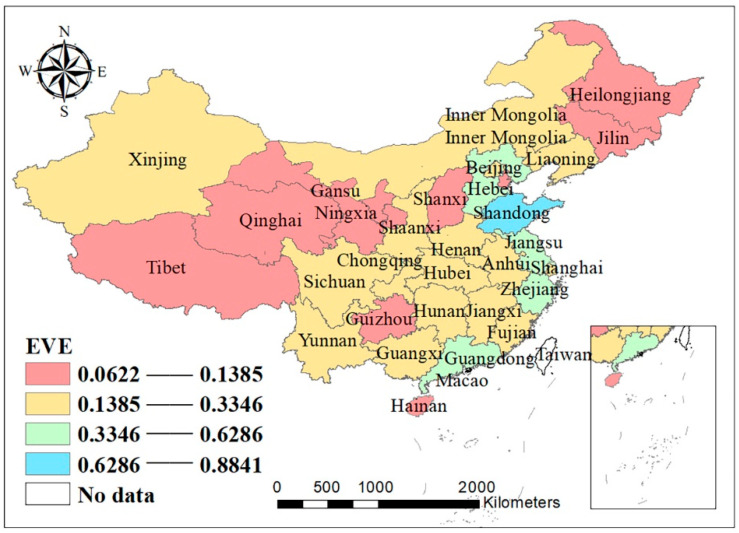
Average values of development indicators of various provinces.

**Figure 2 ijerph-18-12141-f002:**
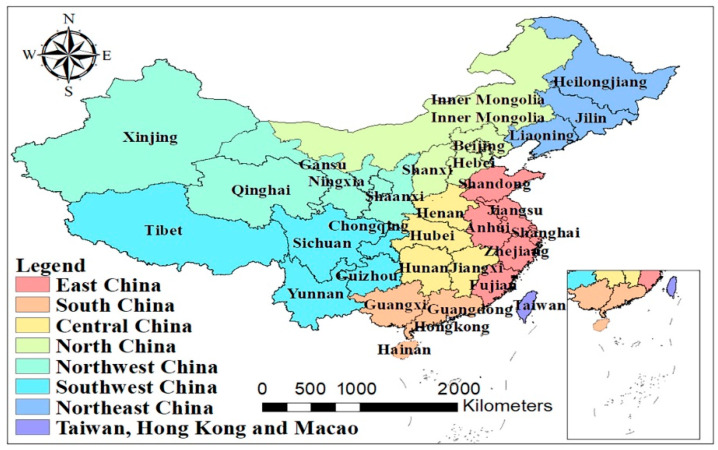
Specific administrative division of China.

**Figure 3 ijerph-18-12141-f003:**
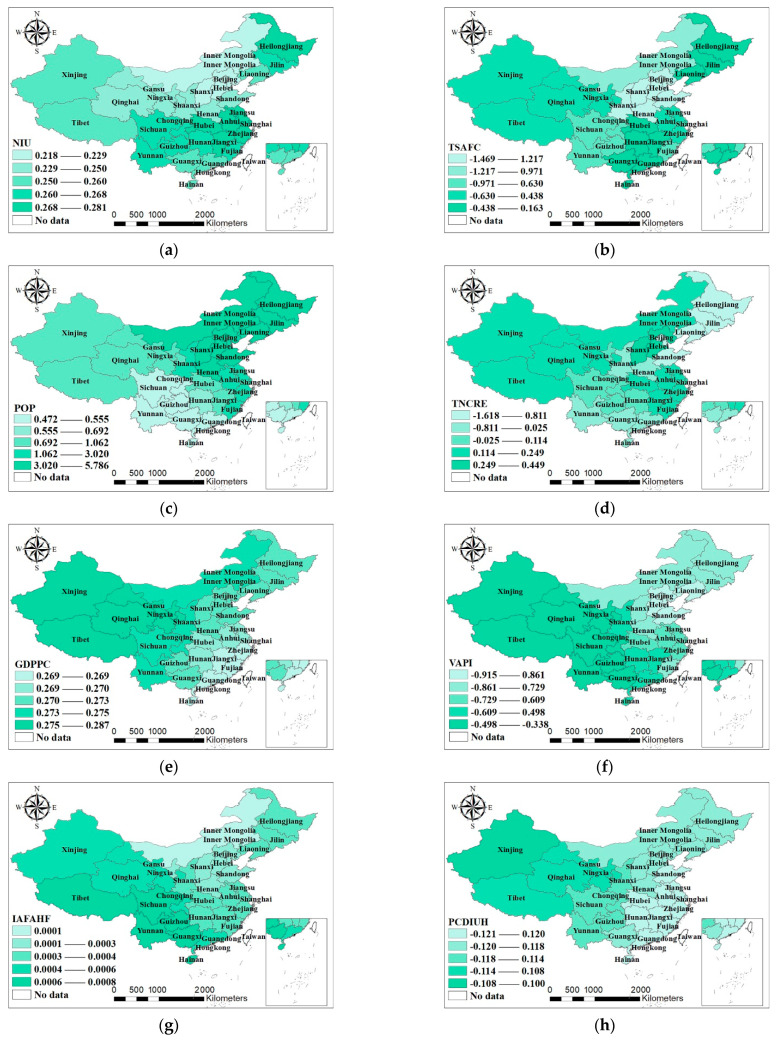
Spatial patterns of coefficients in the MGWR. (**a**) Number of Internet users, (**b**) total sown areas of farm crops, (**c**) population, (**d**) total number of chain retail enterprises, (**e**) GDP per capita, (**f**) value addition of primary industry, (**g**) investment in fixed assets of agriculture, forestry, animal husbandry and fishery, (**h**) per capita disposable income of urban households, (**i**) average relative humidity. (**j**) precipitation, and (**k**) total water resources.

**Table 1 ijerph-18-12141-t001:** Development assessment indicators for the FAPSC.

Primary Indicators	Secondary Indicators	Unit	Nature	References
Development of the FAPSC	Output of fresh agricultural products	10,000 tons	+	[23,24,25]
	Producer Price Index (PPI) of agricultural products	--	−	[26,27]
	Value addition of investment in logistics-related fixed assets	RMB 100 mn	+	[28,29]
	Transaction volume of fresh agricultural products	RMB 10,000	+	[26,30]
	Number of booths in the fresh agricultural product market	--	+	[25,30]
	Number of multi-functional markets selling fresh agricultural products	--	+	[25,30]
	Average retail price index of fresh agricultural products	--	−	[24,26]
	Average Consumer Price Index (CPI) of fresh agricultural products	--	+	[31]

**Table 2 ijerph-18-12141-t002:** Indicators of the influencing factors of the development of the FAPSC.

Primary Indicators	Secondary Indicators	Unit	References
Social environment	Number of Internet users (NIU)	10,000 people	[32]
	Number of employed persons in the primary industry (NEPPI)	10,000 people	[33]
	Total sown areas of farm crops (TSAFC)	1000 hectares	[34]
	Population (POP)	10,000 people	[35,36]
	Total number of chain retail enterprises (TNCRE)	--	[36]
Economic environment	GDP per capita (GDPPC)	RMB	[36,37]
	Value addition of primary industry (VAPI)	RMB 100 mn	[33,36]
	Investment in fixed assets of agriculture, forestry, animal husbandry and fishery (IAFAHF)	RMB 100 mn	[32,33]
	Per capita disposable income of urban households (FCDIUH)	RMB	[37]
Natural environment	Average temperature (AT)	°C	[38,39]
	Average relative humidity (ARH)	%	[39]
	Precipitation (PRE)	mm	[39]
	Sunshine hours (SH)	h	[40]
	Total water resources (TWR)	100 mn m^3^	[40,41]

**Table 3 ijerph-18-12141-t003:** Weighting coefficients of indicators of the FAPSC.

Primary Indicators	Secondary Indicators	Weighting Coefficient
Development of the FAPSC	Output of fresh agricultural products	0.1252
	Producer Price Index (PPI) of agricultural products	0.0237
	Value addition of investment in logistics-related fixed assets	0.0923
	Transaction volume of fresh agricultural products	0.2162
	Number of booths in the fresh agricultural product market,	0.2693
	Number of multifunctional markets selling fresh agricultural products	0.1839
	Average retail price index of fresh agricultural products	0.0446
	Average Consumer Price Index (CPI) of fresh agricultural products	0.0448

**Table 4 ijerph-18-12141-t004:** Average values of development indicators of various provinces’ FAPSC.

Region	Average Value	Region	Average Value
Beijing	0.1718	Hubei	0.2485
Tianjin	0.1325	Hunan	0.2446
Hebei	0.6286	Guangdong	0.4203
Shanxi	0.1241	Guangxi	0.1811
Inner Mongolia	0.1551	Hainan	0.0709
Liaoning	0.2539	Chongqing	0.1799
Jilin	0.1127	Sichuan	0.2125
Heilongjiang	0.1385	Guizhou	0.1161
Shanghai	0.1732	Yunnan	0.1608
Jiangsu	0.4850	Tibet	0.0788
Zhejiang	0.5286	Shaanxi	0.1656
Anhui	0.1994	Gansu	0.1232
Fujian	0.2083	Qinghai	0.0622
Jiangxi	0.1570	Ningxia	0.1085
Shandong	0.8841	Xinjiang	0.1596
Henan	0.3346		

**Table 5 ijerph-18-12141-t005:** Hausman test results.

	Hausman Test
F-statistic	8.7816
Prob	0.0000

**Table 6 ijerph-18-12141-t006:** Multiple linear regression of NEPPI.

Variable	Coef.	Std.Error	t-Statistic	Prob
POP	0.1962	0.0055	35.4027	0.0000
GDPPC	−0.0090	0.0005	−16.9733	0.0000
IAFAHF	0.0059	0.0020	2.9252	0.0037
AT	2.1044	3.0313	0.6942	0.4880
_cons	430.5831	46.8276	9.1951	0.0000

**Table 7 ijerph-18-12141-t007:** Multiple linear regression of TNCRE.

Variable	Coef.	Std.Error	t-Statistic	Prob
POP	1.2360	0.0914	13.5284	0.0000
GDPPC	0.1207	0.0087	13.8223	0.0000
IAFAHF	−0.0687	0.0331	−2.0745	0.0387
AT	289.0761	49.9811	5.7837	0.0000
_cons	−8565.7950	772.1033	−11.0941	0.0000

**Table 8 ijerph-18-12141-t008:** Multiple linear regression of VAPI.

Variable	Coef.	Std.Error	t-Statistic	Prob
POP	0.4138	0.0127	32.5275	0.0000
GDPPC	0.0015	0.0012	1.2654	0.2065
IAFAHF	0.0066	0.0046	1.4401	0.1507
AT	−18.1926	6.9598	−2.0614	0.0093
_cons	104.3503	107.5151	0.9706	0.3324

**Table 9 ijerph-18-12141-t009:** Regression analysis with two-stage least squares (2SLS).

Variable	Coef.	Std.Error	t-Statistic	Prob
NIU	6.26 × 10^−6^	8.49 × 10^−6^	0.74	0.461
NEPPI	1.00 × 10^−4^	0.00	3.19	0.001
TSAFC	−1.87 × 10^−6^	6.26 × 10^−6^	−0.30	0.766
TNCRE	1.67 × 10^−5^	3.15 × 10^−6^	5.30	0.000
VAPI	3.23 × 10^−5^	2.78 × 10^−5^	1.16	0.245
PCDIUH	−2.50 × 10^−6^	1.40 × 10^−6^	−1.78	0.075
ARH	−8.80 × 10^−3^	8.00 × 10^−4^	−10.41	0.000
PRE	1.00 × 10^−4^	0.00	3.90	0.000
SH	4.94 × 10^−7^	3.11 × 10^−6^	0.16	0.874
TWR	−5.18 × 10^−5^	6.89 × 10^−6^	−7.52	0.000
_cons	0.57	0.05	10.48	0.000
R-squared	0.6629			
F-statistic	727.83			

**Table 10 ijerph-18-12141-t010:** GMM regression analysis.

Variable	Coef.	Std.Error	t-Statistic	Prob
NIU	5.19 × 10^−6^	9.04 × 10^−6^	0.570	0.566
NEPPI	1.12 × 10^−4^	0.00	3.910	0.000
TSAFC	−4.43 × 10^−6^	4.18 × 10^−6^	−1.060	0.289
TNCRE	1.53 × 10^−5^	3.18 × 10^−6^	4.800	0.000
VAPI	3.93 × 10^−5^	2.21 × 10^−5^	1.780	0.075
PCDIUH	−1.88 × 10^−6^	1.23 × 10^−6^	−1.520	0.128
ARH	−8.90 × 10^−3^	1.04 × 10^−3^	−8.550	0.000
PRE	1.00 × 10^−4^	0.00	4.960	0.000
SH	5.81 × 10^−7^	1.14 × 10^−6^	0.510	0.611
TWR	−1.00 × 10^−4^	6.39 × 10^−6^	−8.280	0.000
_cons	0.56	0.06	9.590	0.000
R-squared	0.6743			
F-statistic	423.53			

**Table 11 ijerph-18-12141-t011:** Results of the weak instrumental variable test.

Variable	R-Squared	Adjusted R-Squared	Robust F (11,360)	Prob > F
NEPPI	0.9145	0.9119	350.03	0.000
TNCRE	0.7472	0.7394	96.71	0.000
VAPI	0.9064	0.9064	317.1	0.000

**Table 12 ijerph-18-12141-t012:** Results of the over identification test.

	Over Identification Test
F-statistic	1.0917
Prob	0.2961

**Table 13 ijerph-18-12141-t013:** Model indicators of GWR and MGWR.

Model Indicators	GWR	MGWR
R^2^	0.788	0.916
AIC	725.466	278.836
AICc	634.428	313.321
Obs.	373	373
Effective number of parameters	73.9981	70.378

**Table 14 ijerph-18-12141-t014:** MGWR bandwidth.

Indicator	MGWR Bandwidth	GWR Bandwidth
Intercept	90	160
NIU	193	160
NEPPI	53	160
TSAFC	53	160
POP	53	160
TNCRE	53	160
GDPPC	361	160
VAPI	90	160
IAFAHF	361	160
PCDIUH	361	160
AT	67	160
ARH	53	160
PRE	53	160
SH	161	160
TWR	90	160

**Table 15 ijerph-18-12141-t015:** Summary statistics of MGWR parameter estimates.

Indicator	Mean	STD	Min	Median	Max
Intercept	−0.097	0.097	−0.278	−0.108	0.067
NIU	0.255	0.017	0.218	0.259	0.281
NEPPI	−0.197	0.574	−1.226	0.155	0.328
TSAFC	−0.674	0.414	−1.469	−0.573	−0.163
POP	2.361	1.963	0.472	1.674	5.786
TNCRE	0.134	0.559	−1.618	0.073	0.449
GDPPC	0.273	0.004	0.269	0.272	0.287
VAPI	−0.606	0.190	−0.915	−0.609	−0.338
IAFAHF	0.000	0.000	0.000	0.000	0.001
PCDIUH	−0.117	0.005	−0.121	−0.119	−0.100
AT	−0.714	1.188	−3.142	−0.098	0.139
ARH	−0.228	0.307	−0.638	−0.058	0.217
PRE	0.247	0.555	−0.512	0.049	1.517
SH	0.593	0.319	−0.001	0.593	1.089
TWR	0.375	0.268	0.056	0.290	0.921

## Data Availability

The relevant data can be found at the following websites: http://www.stats.gov.cn/, https://data.cnki.net/yearbook/Single/N2020100004, https://data.cnki.net/yearbook/Single/N2020120488, https://data.cnki.net/yearbook/Single/N2020120056, https://data.cnki.net/yearbook/Single/N2020120306.

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
