# Peer review of "Research on the Spatio-Temporal Impacts of Environmental Factors on the Fresh Agricultural Product Supply Chain and the Spatial Differentiation Issue—An Empirical Research on 31 Chinese Provinces"

_ijerph, 2021, doi:10.3390/ijerph182212141_

Round 1

Reviewer 1 Report

Dear authors, thank you for this valuable contribution to agricultural product supply chains. I think it is a relevant contribution and embracing case study from China. Generally, I find the study a bit lengthy, especially due to the presentation of models and formulas. In contrast, I did not see the spatio-temporal aspects reveal so much a) from the study itself, and b) in your conclusions. I would at least grant a section on spatio-temporal impacts in your conclusions. Apart from this general comment, I would have following concrete suggestions:

Line 36. From here on throughout the text, consider using abbreviation for fresh agricultural product supply chain, e.g. “fapsc”, because it reads quite long.

L 55-59: This seems an important contribution of your research that should be highlighted. I suggest to split the sentence and reformulate in order to reveal your objectives better.

Introduction: It is a bit short and should give more background on your study, such as how you conducted it, some details about the case studies, or at least the Chinese context, and a brief overview of how you structure your paper. Mention in that overview the literature review you are starting with.

L 61: Here and in following, use singular for “research”.

L 61-64: Explain shortly how you preceeded with the literature research

L 60 ff., Literature review: Since your literature review is rather brief, I feel it might be better integrated into the introduction, maybe without sub-sections. I think the introduction should end by  114-131. That part is what I feel was the missing part of the introduction.

L 134: Explain features of each model here. L 136-145 can be moved to the end of Section 3. I suggest to add a little discussion of the benefits of each method to the end of Section 3. It could be aspects you already mention under the sub-sections.

L 146: Although you have used the abbreviations od the models earlier, I would write them out in the section headers, e.g. 3.1 “Instrumental Variables Method (IVM)”

L 304: Why is the sown area a social environment factor?

L 397-406: I assume you are drawing here on literature? Please cite references.

L 407: Repetition. Can be removed.

L 459: See L 146.

L 836- 872: Are these conclusions directly drawn from the results of the models? I don’t see them so explicitely in the results section.

Reviewer 2 Report

This paper is about the impact of 14 environmental factors on fresh agriculture food supply chains in the 31 provinces in China during 2008-2019. The authors use four computer models to analyse their environmental impact data and assess which model works best. The aim of this research is important, and the authors write in quite good English, though there are occasional infelicities. However, I would like the authors to respond to the following comments.

First, in the Abstract (lines 21-25), the authors’ statement of their findings is opaque:

The results indicate that: (1) The environmental influencing factors in this paper have significant endogenous problems and various environmental factors impact on the fresh agricultural product supply chain in different trends and to different degrees. (2) With different bandwidth, the environmental factors could impact on the fresh agricultural product supply chain to greatly varied degrees, demonstrating a strong attribute of regional correlation

How can environmental factors have ‘endogenous problems’? Does (1) merely mean that different environmental factors have different impacts? If so, this is not a significant finding but a statement of the obvious. What does ‘different bandwidth’ mean, and how would it increase the variation of environmental impacts? What does ‘a strong attribute of regional correlation’ mean?

Second, on lines 355-356, the authors say: the research covers 31 Chinese provinces, except Hong Kong, Macao and Taiwan, China. This statement implies that Taiwan is a province of China, which is propaganda promulgated by China, and should be deleted from a scientific academic paper.  

I am not qualified to evaluate the elaborate Empirical Analysis set out in section 5, so another reviewer must carry out an assessment of this analysis. If that assessment is favourable, I would recommend publication of the paper, subject to the authors responding to my comments above. The Conclusion in section 6 provides some interesting observations about the implications of the findings, and these observations deserve to be publicised in a respected journal          
